# Indications for Adenoidectomy and Tonsillectomy for Obstructive Sleep Apnea in Children and Adolescents

**DOI:** 10.3390/children13010052

**Published:** 2025-12-30

**Authors:** Boris A. Stuck, Barbara Schneider

**Affiliations:** 1Department of Otorhinolaryngology, Head and Neck Surgery, University Hospital Marburg, Philipps-Universität Marburg, Baldingerstraße, 35043 Marburg, Germany; 2Center for Neuropediatrics and Sleep Medicine, St. Mary’s Children’s Hospital Landshut, Academic Teaching Hospital of the Ludwig-Maximilian Universität, 81377 München, Germany; b.schneider@kinder-schlaf.de

**Keywords:** obstructive sleep apnea, children, adenotonsillar hypertrophy, adenotonsillectomy, tonsillotomy, watchful waiting, polysomnography, pediatric sleep disorders

## Abstract

**Highlights:**

**What are the main findings?**
Snoring and mild forms of obstructive sleep apnea due to adenotonsillar hypertrophy in pre-school and school-age children may be temporary phenomena in this age group and are often self-limiting in nature.On the other hand, children with obstructive sleep apnea may experience a significant impairment in their quality of life and negative effects on their cognitive and emotional development.

**What are the implications of the main findings?**
Watchful waiting may be a treatment option for children with mild symptoms and no other risk factors.The indications for adenotonsillectomy should be based more on clinical assessment and subjective complaints than on the results of polysomnography alone.

**Abstract:**

Obstructive sleep apnea (OSA) in children is a common disorder with significant effects on behavior, cognition, and quality of life. Its diagnosis is primarily based on clinical history and examination, supported by standardized questionnaires such as the Sleep-Related Breathing Disorder subscale of the Pediatric Sleep Questionnaire (SRDB-PSQ), which provides high diagnostic accuracy. Although polysomnography remains the gold standard, its use should be limited to high-risk patients or unclear cases due to availability and cost constraints. Adenotonsillar hypertrophy represents the main cause of pediatric OSA and is often self-limiting. For children with mild symptoms, a watchful waiting approach may be appropriate. Randomized controlled trials (e.g., CHAT, POSTA) demonstrate that spontaneous improvement in polysomnographic parameters occurs in some children, though clinical symptoms often persist. Patients with low apnea-hypopnea-index (AHI), mild obesity, and mild symptoms appear suitable for observation but require a close follow-up. Adenotonsillectomy remains the most effective treatment for clinically significant OSA, leading to marked improvements in sleep quality, daytime symptoms, and quality of life, largely independent of polysomnographic findings. Partial tonsillectomy offers similar efficacy with reduced postoperative morbidity. Management should be individualized and focus on clinical presentation more than on sleep recordings. Future research should focus on identifying which children benefit most from conservative or surgical strategies.

## 1. Introduction

Obstructive sleep apnea (OSA) in children is a relatively frequent disease, affecting children as well as adolescents. The most relevant cause is adenotonsillar hypertrophy, which is, to a certain degree, a physiological state in pre-school and school-age children. Subjective symptoms may be limited and transient in mild cases, but the disease has the potential to significantly affect quality of life and physical well-being. Although polysomnography is the diagnostic gold standard and a prerequisite for the diagnosis of the disease according to the International Classification of Sleep Disorders [1], clinical diagnosis is usually based on medical history including questionnaires and clinical examination only. Tonsillar surgery in terms of (adeno)tonsillectomy (and less frequent partial resection of the tonsils) is usually an effective treatment in children with OSA and tonsillar hypertrophy, although the effectiveness has to be weighed against the natural course of the disease, which often has a self-limiting character. Especially with regard to the associated morbidity of tonsillectomy in children and the limited resources in health care systems, providing the correct indication for tonsillectomy in children with OSA can be challenging. The following narrative review tries to provide an overview of the current indications for (adeno)tonsillectomy in children with OSA and to give recommendations for surgical indications in these cases.

## 2. Materials and Methods

The aim of this article is to discuss the indications for adenoidectomy and tonsillectomy in OSA in children based on the current literature and the personal experiences of the authors. This review does not focus on the management of OSA in children in general, or non-surgical treatment options. Clinical decision-making and indications for tonsillectomy are complex and numerous aspects have to be considered, so the decision for surgical intervention has to be individualized for every affected child. With this regard, an unsystematic database analysis for randomized controlled trials (RCTs), systematic reviews/meta-analyses and clinical guidelines/reviews on treatment options were compiled and discussed in the light of the clinical experience of the authors. In addition, a systematic database analysis in PubMed, Cochrane and Web of Science was performed for articles comparing (adeno)tonsillectomy with watchful waiting using the search terms pediatric, child, children, obstructive sleep apnea, OSA, adenoidectomy, adenotonsillectomy, tonsillectomy, tonsillotomy and watchful waiting. The literature search was restricted to children (age < 18) and articles in English or German published between 01/2000 and 06/2025.

## 3. Results

### 3.1. Obstructive Sleep Apnea in Children and Adolescents

The diagnostic criteria for OSA in children and adolescents according to the American Academy of sleep medicine (AASM) as published in 2014 are as follows [1]:(A)Presence of at least one of the following symptoms:SnoringLabored, paradoxical or obstructive breathing during sleepSleepiness, hyperactivity, behavioral problems or learning difficulties(B)The polysomnographic recording shows one or both of the following abnormalities:○one or more obstructive apneas, mixed apneas or hypopneas per hour of sleep○signs of obstructive hypoventilation, defined as hypercapnia (PaCO2 > 50 mmHG) during at least 25% of total sleep time in conjunction with at least one of the following phenomena:SnoringFlattening of the inspiratory nasal pressure curveParadoxical thoracoabdominal movements

Snoring in childhood is considered the main symptom of a potentially existing OSA. The transitions between simple snoring and OSA are particularly fluid in childhood, and differentiation in clinical practice is not always reliable. A clear definition of snoring in childhood does not exist.

Disease severity based on the AHI is usually classified as follows:

AHI < 1: normalAHI 1–5: mild OSAAHI > 5–10: moderate OSAAHI > 10: severe OSA

For practical purposes, childhood snoring is usually diagnosed when corresponding nocturnal breathing noises are reported by parents or caregivers, but further evaluation reveals no signs of OSA. Formally, the definition of OSA in childhood is based on history and polysomnographic criteria. The indication for therapy, particularly in children, is primarily guided by clinical symptoms.

As already indicated, snoring in childhood may occur episodically and may appear or intensify during an acute infection. In clinical practice, the diagnosis requires symptoms to have been present for a certain time period, for example, four weeks. However, the AASM does not specify such a temporal criterion in the International Classification of Sleep Disorders. Similar temporal variability is also seen in OSA, so that respiratory events often manifest only when, in addition to an individual predisposition, further triggering factors are present, such as an acute infection or infection-related short-term increase in adenotonsillar hypertrophy.

### 3.2. Pathophysiology

The underlying cause of OSA is a functional instability of the upper airway in the segment between the choanae and the trachea. In this collapsible segment, the pharyngeal muscles have to work against the negative inspiratory pressure to maintain airway patency. Various functional and anatomical factors have an impact on this functional instability (see Figure 1). If the patency of the airway can no longer be maintained during sleep, the resistance of the upper airway increases, which initially manifests itself as snoring and can ultimately lead to a temporary closure of the airway in terms of obstructive apneas [2].

In children, anatomical factors leading to a narrowing of the upper airway are, although not exclusively, the main course of this functional instability. Based on the pathophysiological findings, a classification of childhood OSA into three types is proposed in the literature [3,4]:-Type I: adenotonsillar hyperplasia as the main cause-Type II: obesity with only mild lymphoid hyperplasia-Type III: complex craniofacial malformations or neuromuscular diseases

Hypertrophy of the pharyngeal and palatine tonsils is common in childhood and is initially physiological in certain age groups. Depending on the extent, symptoms may be absent or only occur under favorable conditions, such as upper respiratory tract infections. However, pronounced hypertrophy can also lead to manifest OSA. Comorbid factors such as obesity and syndromic diseases can favor the manifestation [5].

Three peaks in frequency can be identified during childhood development. Between the ages of 4 and 5 years there is a significant growth of the pharyngeal tonsils and adenoids, the second peak at the age of 8/9 years is associated with the occurrence of respiratory allergies, and in male adolescents over 15 years of age, testosterone has an influence on the upper respiratory tract [6].
Selection of diseases that cause the occurrence of OSA in children:Obesity, prematurity, cerebral palsy, trisomy 21, Prader–Willi syndrome, Pierre–Robin sequence, craniofacial malformations, dysgnathia, neuromuscular diseases, mucopolysaccharidoses, Chiari 2 malformations, achondroplasia, sickle cell disease


### 3.3. Epidemiology

OSA in children is often a transient phenomenon, the frequency of which depends greatly on age and diagnostic criteria, which are not standardized in various studies. OSA, however, is one of the most common sleep-related breathing disorders in children and adolescents. The main symptom of snoring is found in 34.5% of preschool children [7] and in 6–7% of primary school children [8].

In recent studies, the prevalence of childhood OSA is reported to be 12.8–20.4% in preschool children [9] and 1–4% in older children [10].

However, it should be noted that the prevalence of the disease is significantly higher in corresponding risk groups. For example, based on polygraphic or polysomnographic studies, it can be assumed that up to two thirds of children with trisomy 21 are affected by OSA [11].

### 3.4. Clinical Presentation

The leading symptom of OSA in children and adolescents is snoring. However, the absence of snoring does not rule out the presence of OSA. Instead, parents often report “strained breathing”, “rattling” or a general “increased breathing noise” during sleep. As airway obstruction increases, the snoring noises usually become more intense and signs of hypopnea or apnea appear. Clinical signs of increased work of breathing are jugular or intercostal retractions and paradoxical breathing with opposing movements of the thorax and abdomen during sleep. In order to reduce airway resistance, children often adopt unusual sleeping positions, for example with the head hyperextended. Other typical symptoms include restless sleep with frequent changes of position, night sweats, morning headaches and prolonged or recurrent nocturnal enuresis.

The nocturnal airway obstructions lead to a reduced quality of sleep. The resulting reduction in sleep can manifest itself during the day in the form of behavioral problems, emotional instability, hyperactivity and concentration disorders. They often compensate for their tiredness with increased activity, meaning that hyperactivity or aggressive behavior can also be an expression of a sleep-related breathing disorder. Relevant cardiovascular and metabolic consequences of OSA can also occur in childhood. In addition, growth and developmental disorders can be the result of untreated sleep-disordered breathing. However, the clinical presentation is heterogeneous and age-dependent. Three different clusters of clinical symptoms in pediatric OSA were defined in questionnaire studies [12]:

Cluster 1: nocturnal snoring and daytime sleepinessCluster 2: daytime symptoms with hyperactivityCluster 3: only minor clinical symptoms

Studies show that OSA in children is associated with a measurable reduction in quality of life. It can also cause cognitive deficits and a deterioration in school performance. It is noteworthy that studies have also found an increase in behavioral problems and a tendency towards lower academic performance in children who only snore—without other signs of OSA [13].

### 3.5. Diagnostic Measures

The diagnosis of OSA in children is usually based on medical history, clinical examination and—in selected cases—sleep testing, usually in terms of polysomnography. General medical history and clinical examination of the affected children are mandatory but shall not be explained here in detail. Medical history regarding sleep-disordered breathing has to address the signs and symptoms of OSA as described in the previous chapter on clinical presentation. A standardized medical history and questionnaires can facilitate the collection of medical information and are recommended by the authors. In this context, the subscale for sleep-related breathing disorders of the Pediatric Sleep Questionnaire (PSQ-SRBD Subscale) is of particular importance. It was designed to detect OSA in children and was validated in a cohort of children with confirmed OSA and controls based on polysomnography. Children with mental disorders or severe underlying disease were excluded in the validation studies, so the questionnaire should be used with care in these cases. The test consists of 22 items in four domains with a special focus on respiration, daytime sleepiness and behavior. According to the literature, the PSQ-SRBD subscale has a sensitivity of 81% and a specificity of 87% in the detection of OSA in children. In a later validation study, pre- and postoperative PSG-testing was performed. Although sensitivity and specificity were lower than previously reported, the results correlated well with a postoperative improvement in daytime behavior, indicating that the questionnaire can also be used to assess the outcome of interventions.

In addition to medical history focusing on sleep-disordered breathing, additional signs and symptoms should be assessed that are related to adenotonsillar hypertrophy. Such additional signs and symptoms may have an additional impact on quality of life and may contribute to the indication for treatment, especially for surgery. In some cases, these additional signs and symptoms may be the major driver for the indication for surgery.

Particular attention should be paid to nasal obstruction with mouth breathing, rhinolalia clausa, recurrent upper respiratory tract infections, persistent nasal secretion and hearing impairment (due to chronic otitis media with effusion), which are common symptoms of an enlargement of adenoids in the context of adentonsillar hypertrophy. The time course of these symptoms and previous therapeutic interventions and their effects should be included in the medical history.

Clinical examination should at least include height and weight, and an assessment of upper airway anatomy. Tonsillar hypertrophy can usually be assessed with an inspection of the oral cavity and the oropharynx (see Figure 2). Grading systems for tonsillar hypertrophy, such as the one developed by Brodsky, may be used in this context. When inspecting the adenoids, nasopharyngoscopy (direct or indirect) can be helpful, although it may be challenging to conduct in children. However, a direct visualization of the adenoids is not always necessary and medical history and otoscopy/tympanometry may be sufficient (see below). Attention should be paid to skeletal deformities (e.g., retrognathia), signs of dysgnathia and mouth breathing. In cases of coexisting hearing impairment, an otoscopy and hearing tests (in terms of tympanometry, and, in older children, pure tone audiometry) should be performed. It is important to note that hearing impairment is often not reported by the children themselves but usually suspected by the caregivers. If not spontaneously reported, caregivers should be specifically asked for signs of hearing impairment in children with sleep disordered breathing, as unrecognized hearing impairment can have a negative impact on quality of life and future development. In unclear cases, orthodontists and otolaryngologists should be consulted.

Although objective sleep testing is required to diagnose OSA in children according to the International Classification of Sleep Disorders (ICSD), the role of polysomnography is discussed controversially. Limited capacities and high costs for polysomnography in children have to be considered. To address this, a tailored approach regarding sleep testing based on clinical presentation and medical history is recommended. In children presenting with typical symptoms of OSA and adenotonsillar hypertrophy, treatment (including tonsillectomy) could be initiated without sleep testing, especially if the SRBD-subscale of the PSQ yields pathologic results [14]. Audio or video recordings of sleep, captured by parents or caregivers with a smartphone, can support decision-making. Home sleep testing may be useful in older children but is insufficient to rule out OSA when results are negative [15]. In children with risk factors (e.g., children under the age of 2, with craniofacial abnormalities, syndromic children, children with neuromuscular disease or significant obesity) or in unclear cases, polysomnography should be performed.

### 3.6. Watchful Waiting

Adenotonsillar hypertrophy is a physiological phenomenon in children and is generally self-limiting. Over time, the volume of the adenoids and tonsils typically decreases, which can lead to an improvement in OSA. With this regard, watchful waiting can be an option depending on the clinical presentation and severity of symptoms.

Non-interventional studies regarding the natural course of untreated OSA in children are limited. However, related information can be extracted from randomized clinical trials comparing watchful waiting with various non-surgical and surgical techniques. With respect to the article’s focus, key evidence comes from randomized controlled clinical trials comparing watchful waiting with surgical intervention including only those eligible for adenotonsillectomy.

The ‘CHAT’ trial (Childhood Adenotonsillectomy Trial) included 464 children aged 5 to 9 years with mild to moderate OSA (AHI ≤ 30/h) and compared watchful waiting to surgical intervention after 7 months [10]. Further details of these studies are provided in the following section. In the watchful waiting group, a normalization of polysomnographic findings was observed in 46% of children. The improvement was less pronounced in black children, children with obesity and children with an initial AHI above the median.

Chervin et al. [16] analyzed the results of the watchful waiting group in more detail focusing on clinical parameters. The AHI normalized in 42% of children. However, a partial or complete symptomatic response could be detected in only 15% of children with significant clinical symptoms at inclusion, based on the PSQ-SRBD-Subscale. For a complete response based on polysomnographic parameters, positive predictors included a lower AHI, better oxygen saturation, smaller waist circumference, a higher positioned soft palate, smaller neck circumference, and race other than Black at baseline. A complete spontaneous response was seen in approx. 75% of children with an AHI below 2.5, in approx. 50% of children with an AHI between 2.5 and 4.6 and only in 30% of children with an AHI above 4.6. Predicting factors for subjective improvement were mild symptoms and mild or absent snoring at baseline, absence of witnessed apneas and household smokers, a higher quality of life and being of female sex. The study came to the conclusion that children with lower AHI, a normal waist circumference, less intensive symptoms and less pronounced snoring may be suitable for watchful waiting.

Younger children were included in a randomized controlled study of Fehrm et al. [17] with 53 children at the age of 2–4 years comparing watchful waiting with adenotonsillectomy over a period of 6 months. To be included, children had to present with mild to moderate OSA (obstructive AHI 2–10) and tonsillar hypertrophy (Brodsky score 2–4). In the watchful waiting group, a mean reduction in AHI of 1.9 was detected, although subjective improvement based on the OSA-18 score was limited (56.5 to 52.0). The reduction in AHI was more pronounced in the group with mild OSA based on AHI. At the end of the study, 36% of children in the watchful waiting group received adenotonsillectomy due to persistent symptoms or an elevated AHI in control-polysomnography.

In the most recent study of Redline et al. [18], 459 children at the age of 3 to 13 years with mild OSA (AHI below 3) were included and again watchful waiting was compared to adenotonsillectomy over 12 months. In the primary study endpoints of executive function and attention, only a limited improvement was detected in the watchful waiting group. In the secondary endpoints, a limited improvement in symptoms (including sleepiness) and general and OSA-related quality of life could be demonstrated. In addition, a mild increase in AHI was detected with 13% of children exceeding the study inclusion upper limit of an AHI of 2. Correspondingly, 7% of children presented with moderate OSA (AHI above 5) after the study period.

Based on the results of the randomized studies available, an improvement in OSA can be expected with watchful waiting in a subgroup of children. However, the effects are less pronounced with regard to subjective symptoms and quality of life compared to polysomnographic parameters. In addition, a deterioration over time has to be expected in some children. Children suitable for watchful waiting seem to be less symptomatic and less severely affected with only limited obesity. Nevertheless, disease severity has to be monitored, and children need to be followed carefully.

### 3.7. Surgical Treatment

Apart from surgical interventions for rather specific indications such laryngomalacia, choanal atresia or skeletal deformities, tonsillectomy with or without adenoidectomy is the most frequently performed surgical intervention for OSA in children. Isolated adenoidectomy is often performed in children with symptomatic hypertrophic adenoids and snoring. Because simple snoring and OSA in children are often hard to distinguish and sleep testing is usually not performed before adenoidectomy, children with (mild) OSA are regularly treated with adenoidectomy alone. With regard to the limited evidence available for adenoidectomy for OSA, its efficacy is hard to quantify.

Regarding the effects of adenotonsillectomy, randomized controls trials are available comparing adenotonsillectomy with watchful waiting. In this context, the Childhood Adenotonsillectomy Trial (CHAT) [10] is of particular importance. As mentioned above, this study observed 464 children between 5 and 10 years of age over a period of 7 months. During overnight polysomnography, OSA was defined as an obstructive AHI of 2 or more events per hour (or an obstructive AI of more than 1). Amongst other criteria, children with severe OSA (defined, e.g., by an AHI above 30) were not included with regard to the randomized design including watchful waiting. Although attention or executive function as the primary study outcome was not significantly improved, surgery did reduce symptoms of OSA and improved quality of life and respiratory disturbance according to sleep testing compared to the control group. Compared to baseline, the AHI was reduced in the early adenotonsillectomy group by −3.5, but only by −1.6 in the watchful waiting group. A normalization of the number of respiratory events (defined by an AHI below 2 or an AI below 1) was achieved in 79% of children in the adenotonsillectomy group, but only in 46% of children in the watchful waiting group. Black children, children with high AHI scores at baseline and children with obesity were less likely to show normalized results after 7 months in both groups. With regard to the early tonsillectomy group, 85% of the non-obese children showed normalized results compared to only 67% of the obese children. In addition to the positive effects on polysomnographic parameters, quality of life, as assessed by a series of validated tools, improved significantly more in the adenotonsillectomy group than in the watchful waiting group. Interestingly, according to polysomnography, improvement in OSA severity did not fully explain quality of life improvement [19]. A series of additional secondary analyses was published based on this trial. In one study, an improvement was seen in the Child Behavior Checklist data completed by caregivers for 380 children in the study sample, compared to the watchful waiting group. Interestingly, the positive effect on the various domains of the core (e.g., total Problems score, Internalizing Behaviors, Somatic Complaints, Thought Problems) could be demonstrated at least in part even after removal of sleep-related questions from the analysis [20]. Regarding cardiometabolic parameters, however, a secondary analysis of the study data could not demonstrate a significant change during the study period as a result of adenotonsillectomy [21]. To further assess daytime sleepiness, parent-reported sleepiness was assessed with the modified Epworth Sleepiness Scale for children and the Pediatric Sleep Questionnaire Sleepiness Subscale. Daytime sleepiness was reduced to a greater extent in the surgical group compared to the watchful waiting group. Interestingly, the degree of improvement was only weakly associated with the AHI or with oxygen saturation indices [22]. In addition, a subgroup of 176 children underwent an evaluation of depressive symptomatology. Although an increased risk for depressive symptoms was observed in children with OSA in general and related to oxygen desaturation nadir, symptoms improved over time during the study period, independent of the treatment group [23]. Children undergoing early adenotonsillectomy experienced greater weight gain compared to controls, even in children being overweight at baseline [24]. Adenotonsillectomy, however, was not an independent risk factor for undesirable weight gain [25]. With regard to nocturnal enuresis, a decrease in the prevalence of enuresis was seen in the adenotonsillectomy group but not in the watchful waiting group [26]. Apart from the numerous beneficial effects of adenotonsillectomy, only small effects on cognitive tests were detected [27].

Another large randomized controlled trial was published 10 years later by Redline et al. [27] again comparing adenotonsillectomy with watchful waiting. In this study, 459 children aged 3–13 years were included and followed for 12 months. Apart from the longer follow up period, only children with snoring/mild sleep-disordered breathing were included, defined by an AHI of less than 3. Although changes in executive function and attention were not statistically significantly different between the two groups, children in the adenotonsillectomy group showed greater improvement in behavioral problems, sleepiness and quality of life, and less progression of AHI beyond 3 compared to controls. In addition, after surgery, children had a greater decline in blood pressure levels. In a secondary analysis, it could be demonstrated that adenotonsillectomy was associated with reduced all-cause health care utilization and prescriptions [28]. An increase in body weight was seen in both groups, which also included undesirable weight gain in children already being overweight or obese. However, adenotonsillectomy was not independently associated with undesirable weight gain [29].

Apart from those two studies with large cohorts, a number of smaller studies were conducted with a similar design. Fehrm et al. compared watchful waiting with adenotonsillectomy in a randomized controlled trial in Sweden with 60 children between the age of 2 and 4 years [17]. Children with an obstructive AHI between 2 and 10 were included, based on polysomnography being the primary outcome measure. After six months, obstructive AHI was observed to have decreased in both groups with only a small difference in favor of the intervention group. However, significant differences were seen with regard to the OSA-18 questionnaire as a secondary outcome measure. In addition, a more pronounced difference was seen in the more severely affected children (with regard to AHI). Only 10 of 28 children in the watchful waiting group received adenotonsillectomy after the end of the trial.

The Preschool Obstructive Sleep Apnea Tonsillectomy and Adenoidectomy study (POSTA) included 190 children aged 3–5 years and compared early intervention to watchful waiting (routine waiting list) in children with snoring and mild OSA (AHI ≤ 10) over a period of 12 months [30]. The study focused on cognitive function with global IQ, measured by the Woodcock Johnson III Brief Intellectual Ability (BIA), being the primary outcome measure. From the 141 children completing the study, the BIA data were obtained for 121. No cognitive gain could be demonstrated in the early intervention group compared to the control group. Improvements, however, were seen in the number of respiratory events assessed by polysomnography, sleep quality, parent reported symptoms and aspects of behavior assessed via questionnaires such as the Pediatric Sleep Questionnaire.

Based on these results, adenotonsillectomy leads to a significant reduction in respiratory events and to a significant improvement in daytime symptoms exceeding the effects of watchful waiting. The objective benefit of early surgery with regard to respiratory events is most pronounced in children with a higher degree of OSA based on AHI. Interestingly, the positive effects on daytime symptoms and an improved quality of life exceeded the positive effects on respiratory events. This demonstrates that polysomnography or the AHI as a single measure does not fully reflect the impact of sleep-disordered breathing in children.

It has to be kept in mind that obese children and, e.g., children with Down syndrome are less likely to benefit from surgery, especially the latter showing persistent OSA after surgery in up to 60% [31]. However, these children are also less likely to benefit from watchful waiting.

### 3.8. Complications and Side Effects of Surgical Treatment

Clinical effects of adenotonsillectomy have to be weighed against potential side effects and risks of surgery. While postoperative hemorrhage is the major threat of tonsillar surgery, adenoidectomy is a minimally invasive procedure with negligible surgical risks and rare postoperative bleeding. The rate of postoperative bleeding is a topic of ongoing discussion, and the existing data is limited by the fact that no uniform definition of postoperative bleeding exists in the literature with regard to intensity. Often, postoperative bleeding requiring medical treatment, hospital admission or surgical intervention in particular is meant when rates of postoperative hemorrhage are reported. With this regard, a rate of postoperative hemorrhage of 2.6% was reported in a recent meta-analysis by De Luca et al. [32]. However, although postoperative hemorrhage was divided into primary and secondary hemorrhage, a uniform definition of postoperative bleeding could not be applied. Other reviews also report postoperative hemorrhage rates between 0.1 and 5.7% depending on the type of surgery and the definition of postoperative hemorrhage [33]. Current evidence suggests that postoperative bleeding is more frequent in patients treated for throat infections than in children treated for OSA [32,33].

Apart from postoperative bleeding, respiratory compromise is a frequent postoperative complication and rates of up to 9.4% are reported in meta-analyses, reflecting the fact that children with OSA generally have a higher risk of perioperative respiratory complications. With regard to adenotonsillectomy, children with OSA seem to have a nearly five-times higher risk of postoperative airway compromise compared to children without OSA [32].

A complete description of all potential side effects and adverse events is outside the focus of this article.

### 3.9. Variations of Tonsillectomy

While adenoidectomy is a relatively standardized procedure with little variation in technique, different surgical approaches were developed for the surgery of the palatine tonsils. The complete removal of the tonsils in terms of tonsillectomy can be regarded as the standard of care (although historically partial resections were initially developed before the area of general anesthesia). While various technical options for tonsillectomy exist (“cold steel” dissection being the most widespread), the intervention is defined by the complete removal of the tonsils along their “capsule”. As an alternative approach, partial resection of the palatine tonsils was rediscovered and became more widespread, especially in European countries. In general, partial resection, also often labelled as “tonsillotomy” is associated with less postoperative morbidity and a reduction in postoperative bleeding. In a randomized study from 1999 involving 41 children aged 3.5 to 8 years undergoing either tonsillectomy or tonsillotomy, the latter procedure was associated with less postoperative pain and a faster postoperative recovery [34]. A similar design was used 20 years later with a slightly larger cohort of 79 children aged 2–6 years with similar results [35]. In addition, data from cohort studies suggest that life-threatening or fatal incidences of postoperative bleeding are less frequent with tonsillotomy [36].

On the other hand, a reduced therapeutic effect and the risk for regrowth of the remaining tonsillar tissue has to be considered. Conflicting results were reported in systematic reviews regarding the potential benefits of partial resection [37,38]. Inconsistencies in the terminology of partial resection (“subtotal resection”, “intracapsular resection”, “tonsillotomy”, etc.) and variations in surgical technique (particularly regarding the extent of resection) are limiting factors. In general, the peri- and postoperative morbidity of tonsillar surgery has to be weighed against the expected therapeutic effect. Especially in children with extensive tonsillar hypertrophy (e.g., “kissing tonsils”) subtotal or partial resection is usually effective in treating OSA, making tonsillectomy unnecessary in these cases. In addition, in smaller children with a higher risk for serious courses of postoperative bleeding and when postoperative care cannot be ensured (outpatient surgery, limited availability of emergency management in cases of postoperative bleeding), tonsillectomy should be indicated with care in children with OSA.

## 4. Conclusions/Results

The most relevant cause of OSA in children is adenotonsillar hypertrophy, which is, to a certain degree, a physiological state in preschool and school children. Snoring and mild OSA due to adenotonsillar hypertrophy can therefore be a transient phenomenon in this age group and often has a self-limiting character, which has been demonstrated by various randomized control trials showing an improvement or resolution of symptoms in a relevant number of children with mild to moderate OSA in the watchful waiting group. However, depending on the degree of adenotonsillar hypertrophy, the severity of OSA with regard to AHI and defined risk factors such as obesity, OSA can persist requiring (surgical) intervention, usually in terms of adenotonsillectomy. Early surgical intervention has been shown to significantly improve daytime symptoms including daytime sleepiness, sleep quality, respiratory events and quality of life. These improvements were relatively independent of the improvements in respiratory events during polysomnography. Although sleep testing is required to diagnose OSA in children according to the International Classification of Sleep Disorders, indications for adenotonsillectomy should be based on clinical assessment and subjective complaints rather than on results of polysomnography, as even children with snoring alone can have significant quality of life impairment. In addition, subjective improvement after surgery usually exceeds the effects on polysomnographical findings. Accompanying symptoms of adenotonsillar hypertrophy such as nasal obstruction, mouth breathing, morphometric facial changes and hearing impairment due to chronic otitis media with effusion need to be considered and may contribute to the indication for surgery, as they can lead to long lasting negative consequences for children independent of OSA.

Due to the effects of poor sleep caused by OSA on the cognitive and emotional development of children and adolescents, there is a fundamental need for treatment. Even if watchful waiting is considered as an option, sleep quality frequently remains impaired during this waiting period, with the effects already mentioned. It would also be desirable to better identify the group of people who are unlikely to benefit from surgery, such as children with obesity or Down syndrome. Phenotyping in diagnostics appears to be necessary in order to initiate individualized therapy. Efforts should also be continued to better analyze the complex interactions of various factors in the development of OSA syndrome in childhood in order to be able to counteract its development through preventive measures.

## Figures and Tables

**Figure 1 children-13-00052-f001:**
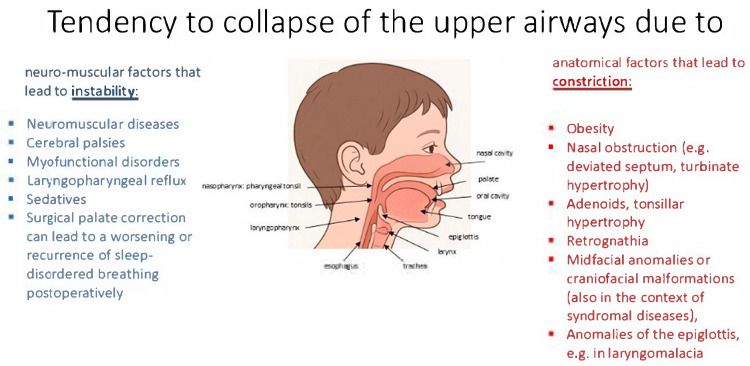
Collapse of the upper airway: risk factors for neuromuscular instability or anatomical obstruction.

**Figure 2 children-13-00052-f002:**
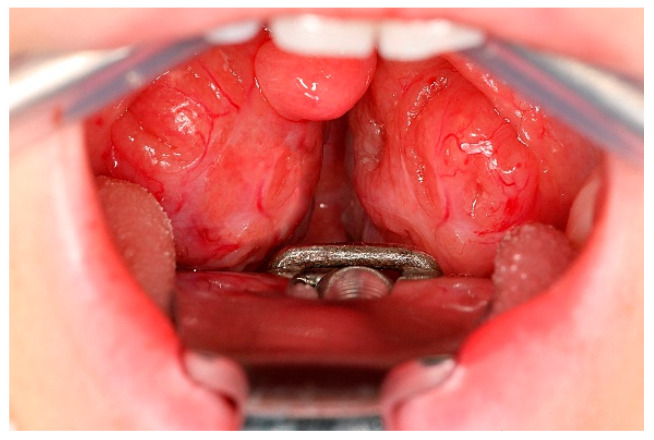
Intraoperative view of enlarged pharyngeal tonsils (intraoperative view; mouth gag and tongue blade in situ).

## Data Availability

Not applicable.

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
