# Peer review of "Indications for Adenoidectomy and Tonsillectomy for Obstructive Sleep Apnea in Children and Adolescents"

_children, 2025, doi:10.3390/children13010052_

Round 1
Reviewer 1 Report
Comments and Suggestions for Authors
This review is fairly written but does not add anything to the known literature. It is written by non-American physicians (but utilizes the American Academy of Sleep Medicine guidelines throughout) and has frequent spelling and word choices that are atypical for a native English speaking audience. In particular, adenotomy is not the common phrase (adenoidectomy more often used) and apnea, not apnoae, should be used throughout. The paper reviews multiple studies on surgery versus watchful waiting for snoring and sleep apnea, but there is no mention whatsoever of medical management. This is a large oversight that must be corrected for a comprehensive review. In addition, "expert opinion" is irrelevant for adenotonsillectomy as expert opinion is best served for rare conditions of procedures, while adenotonsillectomy is one of the most common surgical procedures in children with individual variations offering limited insight into best practices. The sheer number of published studies on adenotonsillectomy and OSA allow a formalized systemic review or meta-analyses of literature, so this review is simply a compilation of studies and findings. The authors only minimally discuss risks of adenotonsillectomy aside from bleeding, which is not quantified, which is a major limitation of the review. The wording throughout has sections of redundancy, particularly when discussing the CHAT trial.
Comments on the Quality of English LanguageWording choices are that of a non-American English speaking writer, while the typical audience for this journal is an American English audience. Frequent mistakes in punctuation noted on page 8, particularly throughout lines 355-401.
Author Response
General remarks: For a special issue, a series of articles regarding obstructive sleep apnea in children has been submitted to “Children”. The presented manuscript is one of the articles of this series. The authors were invited to submit a manuscript specifically addressing “indications for (adeno)tonsillectomy in children with OSA”, as the clinical decision-making process for tonsillectomy is complex and multiple aspects have to be considered. Some of the reviewer would have preferred a more systematic approach rather than the “narrative review” that was submitted (and requested). We agree, that a systematic review considering all the formal aspects and recommendations for such reviews could have been performed. However, such a systematic review would then be necessary for a series of clinical questions addressed in our manuscript: effects of tonsillectomy in comparison to watchful waiting, efficacy and morbidity of tonsillectomy in comparison to partial tonsillectomy, efficacy of adenoidectomy and tonsillectomy in comparison to medical treatment, etc. It seems obvious, that this would have exceeded the limits of such an article. Providing correct indications for surgery has to keep various aspects in mind: age of the children, OSA severity with regard to AHI, severity of symptoms of OSA, additional symptoms of diseases associated with adenotonsillar hypertrophy (e.g., otitis media with effusion, impaired nasal breathing, etc.). We have incorporated aspects of a systematic review for selected aspects of the manuscript (and we have improved this in the revised version), but an article focusing on indication for tonsillectomy in children with OSA has to have an educational character and has be based on clinical experience and expert opinion to a significant degree.
This review is fairly written but does not add anything to the known literature. It is written by non-American physicians (but utilizes the American Academy of Sleep Medicine guidelines throughout) and has frequent spelling and word choices that are atypical for a native English speaking audience. In particular, adenotomy is not the common phrase (adenoidectomy more often used) and apnea, not apnoae, should be used throughout.
We want to thank you for this comment; we have adjusted the language and the wording of the manuscript with regard to American English.
The paper reviews multiple studies on surgery versus watchful waiting for snoring and sleep apnea, but there is no mention whatsoever of medical management. This is a large oversight that must be corrected for a comprehensive review. In addition, "expert opinion" is irrelevant for adenotonsillectomy as expert opinion is best served for rare conditions of procedures, while adenotonsillectomy is one of the most common surgical procedures in children with individual variations offering limited insight into best practices. The sheer number of published studies on adenotonsillectomy and OSA allow a formalized systemic review or meta-analyses of literature, so this review is simply a compilation of studies and findings.
The reviewer refers to the fact that the review does not mention medical treatment and does not discuss the risks of adenotonsillectomy. This is partially correct. However, we want to refer to the fact that this review, being part of a series of publications on OSA in children, specifically addresses “indications for adenoidectomy and tonsillectomy for OSA in children”. This is the focus of this review as expressed in the title and the authors were invited to submit a review specifically addressing indications for surgery. It was not our aim (and not part of the invitation) to review adenotonsillectomy for OSA in children or, moreover, not our aim to review the treatment of OSA in children in general. This is addressed in other articles of the series of manuscripts.
We agree that for a review on the efficacy of adenotonsillectomy for OSA in children, a more systematic review would be required. However, this is not the focus of our review. The aim was to give an overview and recommendation on the clinical decision making, specifically the “indications” for adenotonsillectomy. Indications for surgery are based on the expected effects of the intervention versus the consequences of not performing surgery, which is best expressed by the effects of watchful waiting. Therefore, literature search was focused on randomized controlled trials comparing adenotonsillectomy with watchful waiting. Indications for adenotonsillectomy for OSA in children is a multifactorial decision exceeding the analysis of surgical intervention alone. Therefore, a systematic review on the effects on adenotonsillectomy would not answer the question regarding correct indications for surgery in the clinical setting.
However, we agree that a clearer description of the aim and methodology of the article is required and we have adjusted the manuscript accordingly.
The authors only minimally discuss risks of adenotonsillectomy aside from bleeding, which is not quantified, which is a major limitation of the review.
As described above, the focus of the article is to discuss indications for surgery rather than providing a systematic literature analysis on the effects and side effects of tonsillectomy. However, we agree that risks and side effects of a surgical procedure effect decision making and indications. Therefore, we have addressed side effects and risks of surgery in a more detailed fashion. Nevertheless, a comprehensive description of side effects and risks of adenotonsillectomy is not the focus of this article.
The wording throughout has sections of redundancy, particularly when discussing the CHAT trial.
We have tried to avoid redundancies and reviewed the manuscript critically with this regard.
Wording choices are that of a non-American English speaking writer, while the typical audience for this journal is an American English audience. Frequent mistakes in punctuation noted on page 8, particularly throughout lines 355-401.
We want to thank you for this comment, and we have adjusted the language and the wording of the manuscript with regard to American English and the manuscript has been reviewed by a native speaker.
Reviewer 2 Report
Comments and Suggestions for Authors
This review article presents data on the indications for surgical treatment of obstructive sleep apnea (OSA) in children in the context of adenotonsillar hypertrophy, based on the existing literature. The topic is interesting due to the challenges that may arise in clinical practice when deciding on surgical treatment, considering that watchful waiting may sometimes be sufficient, as adenotonsillar hypertrophy is a physiological phenomenon in children and often self-limiting. The manuscript is well written, aside from a few minor comments:
- Page 2; line 48: Before describing subjective symptoms, I recommend including the main causes of OSA according to the different age ranges in children.
- Page 2; line 74: Please, include the year of the recommendations of the AASM recommendations.
- Page 2; line 79: Polysomnographic criteria should be presented separately and not considered among clinical symptoms.
- Page 3; line 105: I recommend including the AASM severity criteria for OSA in children according to the apnea hypopnea index (AHI).
Author Response
This review article presents data on the indications for surgical treatment of obstructive sleep apnea (OSA) in children in the context of adenotonsillar hypertrophy, based on the existing literature. The topic is interesting due to the challenges that may arise in clinical practice when deciding on surgical treatment, considering that watchful waiting may sometimes be sufficient, as adenotonsillar hypertrophy is a physiological phenomenon in children and often self-limiting. The manuscript is well written, aside from a few minor comments:
- Page 2; line 48: Before describing subjective symptoms, I recommend including the main causes of OSA according to the different age ranges in children.
We have added a statement on the main cause of OSA in the introduction section as requested.
- Page 2; line 74: Please, include the year of the recommendations of the AASM recommendations.
We have included the year of the recommendation of the AASM.
- Page 2; line 79: Polysomnographic criteria should be presented separately and not considered among clinical symptoms.
We want to thank you for this comment. The polysomnographic criteria were initially listed separately, which is in accordance to the AASM recommendations. The error in the listing has occurred in the formatting procedure during submission.
- Page 3; line 105: I recommend including the AASM severity criteria for OSA in children according to the apnea hypopnea index (AHI).
We have included the severity criteria as requested.
Reviewer 3 Report
Comments and Suggestions for Authors< !--StartFragment -->
Dear Authors,
Thank you for the opportunity to review your manuscript. This narrative review addresses an important and clinically relevant topic: the indications for adenotomy and tonsillectomy in children and adolescents with obstructive sleep apnea (OSA). The manuscript presents a well-referenced synthesis of current literature and expert opinion, with practical implications for pediatric sleep medicine and otolaryngology. The following comments are offered to enhance clarity, methodological transparency, and educational value:
Major Comments
- Materials and Methods (Lines 62-70): The manuscript describes a narrative review based on an unsystematic literature search across PubMed, Cochrane, and Web of Science. While relevant search terms and study types are mentioned, the methodology lacks sufficient detail to ensure transparency and reproducibility. Overall, this section would benefit from a more structured and standardized approach to enhance methodological rigor. To strengthen this section, please consider the following:
- Clarify the time frame of the literature search and any language or publication filters applied.
- Specify inclusion and exclusion criteria for study selection.
- Consider referencing established guidelines for narrative reviews, such as SANRA (Scale for the Assessment of Narrative Review Articles), or PRISMA if systematic elements are included.
- Provide a rationale for the selection of key studies (e.g., CHAT, POSTA) and explain how expert opinion was incorporated.
- Scientific Content and Structure: The manuscript is well organized and covers essential aspects of pediatric OSA, including pathophysiology, clinical presentation, diagnostic strategies, and treatment options. The integration of randomized controlled trials (e.g., CHAT, Fehrm et al., Redline et al.) is commendable and supports evidence-based recommendations.
- Consider expanding the discussion on partial tonsillectomy (Lines 422-451), which is mentioned in the Introduction but not elaborated upon in the main text. A brief comparison of outcomes and postoperative morbidity would be valuable.3. Language and Terminology
- Several typographical errors and inconsistencies were noted (e.g., “Narrativ Review and expert statement” in Line 1; “poly-somngraphic” in Lines 351–352). A thorough editorial review is recommended to improve clarity and consistency.
- Long and complex sentences, particularly in Sections 3.5 (Diagnostic Measures, Lines 209–266) and 3.6 (Watchful Waiting, Lines 267–325), could be simplified to enhance readability and accessibility.
- Figures and Visual Aids
- Figure 1 (“Collapse of the upper airway”) should be improved. A clearer schematic diagram with labeled anatomical regions and simplified categorization of risk factors would enhance interpretability.
- Figure 2 (intraoperative view) may benefit from clearer labeling. Alternatively, consider replacing it with a schematic grading of tonsillar hypertrophy to support clinical decision-making.
- Educational Utility: To improve accessibility and practical value for a broader clinical audience, consider adding:
- A summary table stratifying surgical indications by age group, symptom severity, and comorbidities.
- A clinical decision-making flowchart to guide treatment choices (e.g., watchful waiting vs. adenotonsillectomy), based on symptom burden and risk factors.
With major revision, this review has the potential to serve as a valuable resource for clinicians managing pediatric OSA. Thank you again for your contribution to this important area of pediatric sleep medicine.< !--EndFragment -->
Comments on the Quality of English LanguageThe manuscript would benefit from editorial revision to improve clarity and consistency. Several typographical errors and grammatical inconsistencies were noted (e.g., “Narrativ Review,” “poly-somngraphic”), and some sentences are overly long or complex, particularly in the Diagnostic Measures and Watchful Waiting sections. Simplifying sentence structure and standardizing terminology will enhance readability and accessibility for a broader clinical audience.
Author Response
General remarks: For a special issue, a series of articles regarding obstructive sleep apnea in children has been submitted to “Children”. The present manuscript is one of the articles of this series. The authors were invited to submit a manuscript specifically addressing “indications for (adeno)tonsillectomy in children with OSA”, as the clinical decision-making process for tonsillectomy is complex and multiple aspects have to be considered. Some of the reviewer would have preferred a more systematic approach rather than the “narrative review” that was submitted (and requested). We agree, that a systematic review considering all the formal aspects and recommendations for such reviews could have been performed. However, such a systematic review would then be necessary for a series of clinical questions addressed in our manuscript: effects of tonsillectomy in comparison to watchful waiting, efficacy and morbidity of tonsillectomy in comparison to partial tonsillectomy, efficacy of adenoidectomy and tonsillectomy in comparison to medical treatment, etc. etc.. It seems obvious, that this would have exceeded the limits of such an article. Providing correct indications for surgery has to keep various aspects in mind: age of the children, OSA severity with regard to AHI, severity of symptoms of OSA, additional symptoms of diseases associated with adenotonsillar hypertrophy (e.g., otitis media with effusion, impaired nasal breathing, etc.). We have incorporated aspects of a systematic review for selected aspects of the manuscript (and we have improved this in the revised version), but an article focusing on indication for tonsillectomy in children with OSA has to have an education character and has been based on clinical experience and expert opinion to a significant degree.
Materials and Methods (Lines 62-70): The manuscript describes a narrative review based on an unsystematic literature search across PubMed, Cochrane, and Web of Science. While relevant search terms and study types are mentioned, the methodology lacks sufficient detail to ensure transparency and reproducibility. Overall, this section would benefit from a more structured and standardized approach to enhance methodological rigor. To strengthen this section, please consider the following:
Clarify the time frame of the literature search and any language or publication filters applied.
We have clarified our literature search strategy as requested.
Specify inclusion and exclusion criteria for study selection.
We have clarified our literature search strategy as requested.
Consider referencing established guidelines for narrative reviews, such as SANRA (Scale for the Assessment of Narrative Review Articles), or PRISMA if systematic elements are included.
Provide a rationale for the selection of key studies (e.g., CHAT, POSTA) and explain how expert opinion was incorporated.
Systematic elements were incorporated with regard to the literature search for RCTs comparing (adeno)tonsillectomy with watchful waiting. PRISMA was not incorporated as the number of RCTs was low and basically all available studies were considered. With regard to the complex decision-making process, where efficacy of tonsillectomy is only one part that has to be considered, we tried to explain the various aspects that have to be kept in mind when discussing surgical indications. For such a multifactorial decision-making process, external evidence is limited but internal evidence and clinical experience is requested. This is where the authors incorporated the expert opinion.
Scientific Content and Structure: The manuscript is well organized and covers essential aspects of pediatric OSA, including pathophysiology, clinical presentation, diagnostic strategies, and treatment options. The integration of randomized controlled trials (e.g., CHAT, Fehrm et al., Redline et al.) is commendable and supports evidence-based recommendations.
Consider expanding the discussion on partial tonsillectomy (Lines 422-451), which is mentioned in the Introduction but not elaborated upon in the main text. A brief comparison of outcomes and postoperative morbidity would be valuable.
In the chapter mentioned, partial tonsillectomy was described and a brief comparison of outcomes and postoperative morbidity is already provided, referring to the most relevant clinical studies in the field. With regard to the fact that “indications for surgery” is the main focus of the article, we believe that partial tonsillectomy is sufficiently discussed.
Several typographical errors and inconsistencies were noted (e.g., “Narrativ Review and expert statement” in Line 1; “poly-somngraphic” in Lines 351–352). A thorough editorial review is recommended to improve clarity and consistency.
Long and complex sentences, particularly in Sections 3.5 (Diagnostic Measures, Lines 209–266) and 3.6 (Watchful Waiting, Lines 267–325), could be simplified to enhance readability and accessibility.
We want to thank you for this comment. We have corrected the two sections mentioned and we have adjusted the language and the wording throughout the manuscript with regard to American English and the manuscript has been reviewed by a native speaker.
Figures and Visual Aids
- Figure 1 (“Collapse of the upper airway”) should be improved. A clearer schematic diagram with labeled anatomical regions and simplified categorization of risk factors would enhance interpretability.
- Figure 2 (intraoperative view) may benefit from clearer labeling. Alternatively, consider replacing it with a schematic grading of tonsillar hypertrophy to support clinical decision-making.
Figure 1 has been revised, the graphic has been supplemented with anatomical terms, and the classification of risk factors for neuromuscular instability or anatomical obstruction as causes of upper airway obstruction has been clarified.
A legend has been added to figure 2 to clarify its content.
Educational Utility: To improve accessibility and practical value for a broader clinical audience, consider adding:
- A summary table stratifying surgical indications by age group, symptom severity, and comorbidities.
- A clinical decision-making flowchart to guide treatment choices (e.g., watchful waiting vs. adenotonsillectomy), based on symptom burden and risk factors.
We completely understand the idea behind this aspect. Nevertheless, one thing we wanted to make clear in our manuscript is that the indication for surgery is a multifactorial decision, involving numerous aspects such as age, objective disease severity, severity of symptoms, clinical presentation, anatomical condition, comorbidities and associate symptoms and diseases that come along with adenotonsillar hyperplasie in children, apart from OSA. We do not believe that this can be simplified using flowcharts or tables.
The manuscript would benefit from editorial revision to improve clarity and consistency. Several typographical errors and grammatical inconsistencies were noted (e.g., “Narrativ Review,” “poly-somngraphic”), and some sentences are overly long or complex, particularly in the Diagnostic Measures and Watchful Waiting sections. Simplifying sentence structure and standardizing terminology will enhance readability and accessibility for a broader clinical audience.
We want to thank you for this comment and we have adjusted the language and the wording of the manuscript with regard to American English and the manuscript has been reviewed by a native speaker.
Round 2
Reviewer 1 Report
Comments and Suggestions for Authors
The authors clarify this was a requested narrative review comparing surgery and watchful waiting, but this is never stated within the text, nor is there a "limitations" section to clarify this. It still reads as a review that leaves a lot out when considering surgery in children and does not add significantly to the existing literature. The article revision is certainly improved, but I do not feel it provides sufficient information for an acceptance. I leave it to the editors to determine if you fulfilled their needs.
Author Response
We understand that the reviewer does not appreciate the general concept of the manuscript, although we explained the methodology in the methods section more clearly. However, we have exactly provided what were were asked for by the editor of the special issue, so we regret that we cannot change the overall concept of the manuscript.
Reviewer 3 Report
Comments and Suggestions for Authors
Thank you for your careful revisions. The manuscript has improved substantially and now provides a balanced, clinically relevant overview of the indications for adenotomy/adenoidectomy and tonsillectomy in children and adolescents with obstructive sleep apnea. The integration of randomized controlled trial evidence (CHAT, POSTA, Fehrm, Redline) alongside clinical experience strengthens the narrative and makes the review useful for pediatric otolaryngologists, sleep specialists, and pediatricians. Congratulations!
Author Response
We want to thank the reviewer for this enthusiastic response.